# *Mycobacterium abscessus* Virulence Factors: An Overview of Un-Explored Therapeutic Options

**DOI:** 10.3390/ijms26073247

**Published:** 2025-03-31

**Authors:** Mario Cocorullo, Alessandro Stamilla, Deborah Recchia, Maria Concetta Marturano, Ludovica Maci, Giovanni Stelitano

**Affiliations:** Department of Biology and Biotechnology “Lazzaro Spallanzani”, University of Pavia, 27100 Pavia, Italy; mario.cocorullo01@universitadipavia.it (M.C.); alessandro.stamilla@unipv.it (A.S.); deborah.recchia@unipv.it (D.R.); mariaconcetta.marturano@unipv.it (M.C.M.); ludovica.maci@unipv.it (L.M.)

**Keywords:** *Mycobacterium abscessus*, virulence factors, GPL, TTP, iron uptake, secretion systems, efflux pumps, biofilm

## Abstract

*Mycobacterium abscessus* (*Mab*) is an opportunistic pathogen gaining increased importance due to its capacity to colonize the respiratory tract of patients with chronic lung diseases such as individuals with Cystic Fibrosis. The actual therapeutic regimen to treat *Mab* infections is based on repurposed drugs from therapies against *Mycobacterium tuberculosis* and *avium*. In addition to the need for new specific drugs against this bacterium, a possible strategy for shortening the therapeutic time and improving the success rate could be targeting *Mab* virulence factors. These drugs could become an important integration to the actual therapeutic regimen, helping the immune system to fight the infection. Moreover, this strategy applies a low selective pressure on the bacteria, since these elements are not essential for *Mab* survival but crucial for establishing the infection. This review aims to provide an overview of the *Mab*’s virulence factors that are poorly studied and mostly unknown, suggesting some interesting alternatives to classical drug development.

## 1. Introduction

The *Mycobacterium* genus includes species that exhibit significant phenotypic and genotypic diversity. They are grouped into tuberculous mycobacteria (TB), mycobacteria causing leprosy, and non-tuberculous mycobacteria (NTM) which are environmental species with different features [1]. For instance, NTM include rapid and slow growing mycobacteria such as the rapidly growing *Mycobacterium chelonae-abscessus* complex (MABSC), which includes *M. abscessus* subsp. *abscessus*, *M. abscessus* subsp. *bolletti*, *M. abscessus* subsp. *massiliense*, *M. chelonae* and *M. fortuitum* or the slow-growing *M. avium* complex (MAC) [2]. Although NTM exposure is prevalent in humans, the majority of NTM have minimal pathogenicity. However, NTM infections are becoming a growing worldwide health problem due to the prevalence of NTM in both natural and man-made environmental niches combined with host risk factors [3]. Clinical signs and symptoms may include chronic lung disease, disseminated disease, and skin infections.

*Mycobacterium abscessus* (*Mab*) was considered for years as a subspecies of *Mycobacterium chelonae*, but genetic studies highlighted a well-characterized bacterial species [4]. Its genome consists of a circular chromosome of 5,067,172 base pairs including 4920 predicted coding sequences (CDS) rich in G + C for 64%. Within the genome, a 23 kb mercury resistance plasmid and five insertion sequences were detected, unlike other sequenced mycobacteria [5].

*Mab* has emerged as an important human pathogen over the last 10 years. Indeed, it has the capability to colonize the lung airways of patients with cystic fibrosis (CF), chronic obstructive pulmonary disease (COPD), or bronchiectasis [6]. Similarly to *M. tuberculosis* (*Mtb*), *Mab* is an intracellular bacterium capable of growing in macrophages and free-living amoebas [7]. Moreover, it may form biofilm in soft tissues that are hard to eradicate [8].

*Mab* colonies may be smooth (S) or rough (R) showing major differences in their pathogenic profile [5]. Compared to S, the R morphotype lacks the glycopeptidolipids (GPL), a surface polyketide compound. So, the switch from S to R is due to the loss of GPL [5]. The R form is associated with more severe infections in mice, and it is the only form able to persist and multiply in a murine pulmonary infection model [4,5].

*Mab* is resistant to many drugs thanks to a plethora of mechanisms including its thick cell wall, the efflux pumps and metabolic enzymes [3]. Its antimicrobial resistance (AMR) may also be acquired from the environment or adaptive due to point mutations that can alter the susceptibility to drugs [9]. Therefore, only a small number of effective antibiotics are available for *Mab* treatment, most of which are repurposed from a therapeutic regimen originally optimized against *Mtb* and *avium* [10]. The recommendations from the American Thoracic Society/Infectious Diseases Society of America include macrolides (typically Azithromycin), Aminoglycosides (Amikacin), Carbapenems (Imipenem), and Cephamycins (Cefoxitin) [2]. Nevertheless, anti-*Mab* treatments have relatively low success rates ranging between 33% and 57% depending on the strain [10]. The first specific inhibitor of cell division highly effective against *Mab* and considered a promising putative drug has been recently developed [11].

A possible alternative to fight *Mab* infections could be targeting its virulence factors. This strategy aims to block the establishment of infection instead of killing the bacteria, applying a low selective pressure thus avoiding the insurgence of resistance [12]. The integration of drugs targeting *Mab*’s virulence factors into the actual therapeutic regimen will enhance the outcome, possibly shortening the time and improving the patient’s quality of life. This strategy has been successfully applied to combat other bacterial infections, such as *Pseudomonas aeruginosa* and *Staphylococcus aureus*, leading to the development of new drugs and vaccines, some of which are already in clinical trials [13,14]. For instance, the AstraZeneca compound MEDI4893 is an antibody targeting the α-toxin of *S. aureus* in phase I clinical trials [14].

The current review aims to provide an update on the recent discoveries of *Mab*’s virulence factors as putative targets and provides new ideas for the development of novel drugs against *Mab*.

## 2. Overview of Known *M. abscessus*’ Virulence Factors

Many key aspects of *Mab*’s capability to replicate inside the macrophages are still unknown. Like many other intracellular pathogens, *Mab* utilizes multiple receptors to enter the host cell, including complement receptors and mannose receptors [15]. Once in the phagosome, *Mab* arrests its maturation by inhibiting the phagosome-lysosome fusion, surviving inside this organelle which is its replicative niche [16]. However, this mechanism is poorly understood and probably involves the modulation of host cell signaling pathways [17]. Within the host cells, *Mab* is protected from host immune responses, antibiotic treatments, and certain antimicrobial peptides, as intracellular bacteria are harder to eliminate than extracellular ones. Through this strategy, *Mab* can persist in the host for extended periods, leading to chronic infections that are challenging to treat [18].

*Mab* produces a range of virulence factors that drive immune evasion and intracellular survival, including various lipids, proteins, and enzymes. For example, the GPL plays a key role in biofilm formation and immune modulation. Additionally, *Mab* secretes enzymes like catalases and superoxide dismutases to neutralize the reactive oxygen species produced by host cells to kill the bacteria [19]. Most of *Mab*’s virulence factors are typical of the genus *Mycobacterium*. However, clusters of genes that are not present in other mycobacteria have been found in *Mab*’s genome, probably acquired by horizontal gene transfer. Some of these encode for virulence factors while others for drug-inactivating enzymes [20].

### 2.1. Surface Elements

The presence of GPL on the *Mab* cell membrane influences its hydrophobicity and potentially affects bacterial adhesion and virulence, conferring colonies with different appearances [21]. S colonies expressing GPL appear round and glossy, while R morphotypes, lacking GPL, have a dry, corded appearance [22]. The transition from S to R is typically the result of mutations in genes within the GPL locus that are involved in the synthesis and secretion of this molecule [23]. Although the exact mechanisms behind this switch remain unclear, stress conditions encountered during the host colonization may drive the transition from S to R. Often, the R morphotype is isolated from patients with CF having chronic pulmonary infections. Furthermore, it is associated with heightened virulence compared to S morphotypes [24].

Post-synthesis modifications of GPL may influence *Mab* adhesion to and invasion of macrophages. Changes in GPL expression have been observed when the bacteria grow in artificial CF sputum [25,26]. The mechanical implications of the morphotypes are quite fascinating: while the S generally exists as a single bacilli, the R aggregates in bacterial clumps that phagocytic cells struggle to destroy [27]. Moreover, the S morphotype survives in a poorly acidified phagosome manipulating the macrophage functions to avoid the phagolysosome fusion [27]. In contrast, the R morphotype clumps may survive within the mature phagolysosome as growing aggregates, which leads to the mechanical disruption of this organelle or the formation of autophagic vacuoles, ultimately resulting in macrophage apoptosis [20,28]. Following, the R bacilli are released in the extracellular space where they express large serpentine cords acquiring resistance to further phagocytosis and provoking significant inflammation and abscess formation [20].

The trehalose polyphleates (TPP) consist of a trehalose core esterified by different polyunsaturated fatty acids known as phleic acids [29]. These high-weight glycolipids are not toxic, do not cause the release of proinflammatory cytokines nor prevent the phagolysosome fusion. In particular, *Mab* shows a high content of octoacylated trehalose with seven phleic-acid-like fatty acid substituents and a C_14_-C_19_ fatty acid residue [29]. Indeed, the synthesis of the TPP begins in the cytoplasm where a 2,3-diacyl trehalose precursor is substituted with the C_14_-C_19_ fatty acyl group [29]. This molecule is externalized by MmpL10 on the bacterial cell surface, where the acyl transferase PE catalyzes the trans-esterification of a C_36_-C_40_ phleic acyl substituent to generate the mature TPP [30].

TPP has been identified as a key element of the R colonies surface responsible for aggregation [31]. It is the main component of cords, serpentiform structures formed through end-to-end and side-to-side aggregation of bacilli, where the long axis of the cording is parallel to the orientation of the long axis of each cell [30]. TPP plays a major role in overwhelming the bactericidal capability of macrophages via the formation of bacterial clamps that engulf the phagosome [31]. Notably, the bacteriophages BPs, Muddy, and other TPP-dependent phages are required to recognize and bind TPP, probably as a coreceptor, to infect *Mab* and *M. smegmatis* [32].

The biosynthesis of the cell membrane and wall is regulated by different enzymes catalyzing the formation of GPL, TPP, mycolic acids, and other components. CyCs are Cyclipostins and Cyclophostin analogs inhibiting *Mtb* enzymes involved in this pathway, potentially exhibiting a multitarget behavior [33]. Recently they have been evaluated against *Mab*, too. Experimental data suggests that CyCs inhibit two putative β-lactamases Rv1730c and Rv1367c, the serine hydrolase Rv0554, five members of the lipase family Lip of which Rv1399c and Rv3203 have been characterized as LipH and LipV respectively, six enzymes with lipolytic activity such as the monoacylglycerol lipase Rv0183, the esterase Rv0045c and the antigenic Ag85a (Rv3804c) and Ag85c (Rv0129c) proteins [33]. Inhibition of these enzymes showed a toxic effect on *Mab* cells, also inhibiting their virulence [34]. In several studies, CyC_17_ has also been shown to be the most potent inhibitor among the CyCs, displaying minimum inhibitory concentration (MIC) values comparable to most antibiotics used in therapy against *Mtb* and *Mab* infections [34,35]. This data underlines the attractivity of this new family of compounds as drug candidates for future therapeutic use against mycobacterial-associated infections, particularly against *Mab* [34,35]. This multitarget strategy could be an interesting integration into the current therapeutic regimen, as bacteria need simultaneous mutations in all targets to acquire resistance to a single molecule [36]. However, this would occur with low frequency due to the non-essentiality of some targets [34].

Polar lipids are also involved in *Mab*’s resistance to human defenses. The antimicrobial peptide cathelicidin LL-37 is a linear cationic-α-helical structure. It is produced by immune cells in response to pathogenic microbes and plays a crucial role when inflammation occurs [37]. Despite LL-37 is effective against *M. tuberculosis*, it shows no activity against NTMs [38]. This lack of activity is possibly related to polar lipids and their derivatives, such as cardiolipin or phosphatidylinositol mannosides, as the GPL composition does not affect *Mab*’s resistance to this peptide [38,39]. These findings highlight the potential of inhibiting lipid biosynthesis as a strategy to combat *Mab*, both directly and indirectly.

Another mechanism involved in *Mab* virulence is related to modifications of surface elements. For instance, lipoprotein acylation and glycosylation increase the resistance to antibiotics and reactive oxygen species (ROS) by decreasing membrane permeability and to lysozyme by masking the peptidoglycan [40]. This has been demonstrated through the knockout of *MAB*_1122c codifying the protein-O-mannosyltransferase Pmt. Pmt is responsible for the glycosylation of certain lipoproteins, such as SodC, that are part of the mycobacterial cell envelope. The increased cell wall permeability of Pmt knockout strains leads to an increased susceptibility to innate cellular defenses, such as ROS [40]. The modification of membrane permeability in order to increase the drug delivery to pathogenic bacteria may be an interesting strategy to take into account in drug design and a valid addition to the actual therapeutic regimen.

### 2.2. Secretion Systems and Efflux Pumps

Mycobacteria have three different secretion systems: the Twin-Arginine Translocation (TAT) pathway, the Sec pathway, and the type VII secretion system, also known as the ESX secretion system.

TAT is an essential secretion system that enables the translocation of polypeptides in the folded state which is not present in all bacteria [41]. All substrates exported by TAT share a highly conserved two-arginine leader motif located in the N-terminus of the protein, typically two arginine followed by a generic amino acid and two hydrophobic residues [41,42]. Three membrane proteins contribute to form the TAT export system in *Mtb*: TatA, TatB, and TatC. A characteristic of this system is that during protein export, oligomeric TatA is enrolled in the TatB-TatC protein complex giving rise to the translocase channel [43]. TAT is involved in the secretion of virulence factors such as the phospholipase C PlcB, a known TAT-mediated secretory-virulent protein, in pathogenic mycobacteria [31].

Sec allows the export of different substrates through the cytoplasmic membrane. Two non-redundant Sec systems have been identified in mycobacteria: SecA1 which is an essential housekeeping system and SecA2, an accessory secretion factor. SecA1 substrates are precursor proteins containing a conserved amino-terminal signal sequence. Upon translocation, the signal peptide is cleaved to generate the mature exported proteins. For example, a family of serine-rich repeat (SRR) glycoproteins functioning as adhesins is exported by Sec [44]. SecA2 secretes a smaller pool of substrates lacking a signaling sequence, mainly virulence factors such as SodA, SapM, and PknG [45,46] Similarly, the Sec system secretes the kinase PknG and the esterase LipO, both of which contribute to blocking the phagosome maturation [46,47].

The early secretory antigenic target (ESAT-6) secretion systems, known as Type VII secretion systems (ESX), are typical of mycobacteria and play a key role in facilitating protein transport across the outer membrane, significantly contributing to *Mab* virulence but also to nutrient acquisition and bacterial conjugation [48]. ESX shares a conserved set of genes among mycobacterial species. These genes codify the machinery required for substrate secretion, accessory proteins with different functions, and the ESX-secreted proteins [49]. In *Mab*, three secretion systems have been characterized: ESX-3, ESX-4, and ESX-P (Figure 1). ESX-3, composed of EsxG/H proteins, induces pro-inflammatory cytokines production in bone marrow-derived macrophages. Mice infected with ESX-3-deleted *Mab* show reduced inflammatory and granulomatous responses along with lower bacterial survival [50]. However, the specific mechanisms of ESX-3 promoting *Mab* growth remain unclear. ESX-3 role related to the iron-uptake can be studied in iron-rich and iron-limited environments, modulating differently *Mab* growth [51]. Under iron-limited conditions, ΔESX-3 mutants struggle to grow, causing suppression of the activity of succinate dehydrogenase (SDH), a key iron-dependent enzyme in the tricarboxylic acid (TCA) cycle, resulting in dysregulation of the TCA cycle [51]. Further studies have shown that the regulator MtrA is able to bind ESX-3 and thus enable both extracellular and intracellular survival of *Mab* [51].

Similarly, the lack of the ESX-4 locus decreases the survival of *Mab* within macrophages and amoebae. This is due to the effect of the ATPase EccB4, associated with this secretion system, that enhances *Mab* virulence by limiting the phagosome acidification and inducing its rupture followed by the bacterial migration to the cell host cytosol [50]. *Mab*’s ESX-4 system, functionally analogous to the role of ESX-1 in *Mtb*, is also composed of the EccE component, a membrane- and cell wall-associated protein, and the two main effector proteins EsxU and EsxT forming a 1:1 heterodimer with the ability to permeabilize artificial membranes [52]. The secretion of this dimer is restricted in eccB4-deletion mutants that show a lower efficiency in blocking the acidification and inducing damage to the phagosome. These mutants are also associated with a decreased interaction of *Mab* with the macrophage cytosol [52]. Notably, the lack of esxU/esxT is associated with a hypervirulent strain, suggesting that other effector proteins may be involved in pathogenesis possibly exploiting the ESX-4 secretion system [53]. These results confirmed the essential role of ESX-4 in the intracellular behavior of *Mab*.

In a recent work, a library of *Mab* mutants revealed the small regulatory RNA (sRNA) B11 as necessary for the smooth morphology of *Mab* [54]. The B11 deleted mutant was constructed and its gene expression and phenotypes associated with pathogenesis were characterized. The loss of B11 resulted in increased virulence and pro-inflammatory immune communication, as well as altered expression of many genes of which the complementary sequences to B11 in their ribosome binding sites (RBS) were enriched [54]. Among the genes overexpressed, several components of the ESX-4 system were identified, one of which was found to be repressed by direct binding of B11 to its RBS. This study demonstrated that mutations that reduce the B11 activity are beneficial for bacteria in certain clinical contexts [54].

Other studies identified additional plasmid-borne ESX components in *Mab* clinical isolates, suggesting a more extensive presence of the ESX systems in this species than previously recognized, potentially representing a novel pathway through which *Mab* acquires virulence factors that drive evolving pathogenicity [55].

The mycobacterial membrane protein large (MmpL) are efflux pumps playing an essential role in host-pathogen interaction. This includes the export of complex lipids that are crucial for maintaining the integrity of the mycobacterial envelope, the first barrier against immune cells and chemotherapeutic agents [56]. *Mab* possesses 31 different MmpL proteins compared to the only 13 of *M. tuberculosis*, underlining their importance for fast-growing mycobacteria [57]. For instance, MmpL4 transports the GPL to the outer mycobacterial surface. Disrupting MmpL4 causes a shift from S to R morphotype, which is associated with increased virulence through the formation of the large serpentine cords, triggering pro-inflammatory responses and apoptosis [58]. Similarly, the disruption of MmpL8 reduces the transport of glycosyl-diacylated-nondecyldiols, comprising different combinations of oleic and stearic acids, and is responsible for the bacterial adhesion to macrophages, inducing a delayed macrophage phagosome rupture therefore lowering the mycobacterial virulence [59]. The MmpL8 *Mab* KO mutant capable of blocking phagosome acidification, like the WT strain, shows a delay in damaging the phagosome membrane and contacting the cytosol [59]. This mutant was used in vivo in zebrafish models, and the mutant’s attenuation of virulence was found to be confirmed by an altered in vivo killing of zebrafish and a reduced propensity to induce granuloma formation [59].

Ongoing studies focus on gaining a deeper knowledge of *Mab*’s efflux pumps and secretion systems. The obtained knowledge will be of great help in designing new drugs to block these systems, with consequential loss of virulence for pathogenic strains. An alternative strategy could exploit the RNA silencing to block the expression of secretory systems at the transcriptional level. In the near future, this novel methodology may revolutionize the fight against pathogenic species.

A special mention is required for the secreted virulence factors of pathogenic mycobacteria that are still not studied in *Mab* but are known in other mycobacteria. Proteins such as the phosphatases PtpA, PtpB, and SapM play a major role in *Mtb* virulence and pathogenicity, playing a key role in blocking phago-lysosome maturation [60]. The inhibition of the homologous *Mab* phosphatases could become a winning strategy to impair *Mab* infection.

Another important secreted protein is the complex Ag85 (MAB_0177), composed of the paralogous subunit A (MAB_0176), B (MAB_1579), and C (MAB_0175), with a double function that has been identified in *Mtb*. Ag85 remains attached to the bacterial surface after secretion, where it is responsible for the biosynthesis of the trehalose dimycolate and mycolylation of arabinogalactan in the final stage of cell wall assembly by using trehalose monomycolate as a substrate [34,35,61,62]. The second function is related to the adhesion to the host cell, the first step of invasion during infection. Experimental data suggests that Ag85 can bind the fibronectin, the main component of the host extracellular matrix, through the unique binding sequence ^98^FEWYYQSGLSV^108^ present on Ag85B and mapped on *Mycobacterium kansaii* [61]. CyC_17_ is reported in the literature as a promising inhibitor of the Ag85A and Ag85C catalytic activities. An alternative strategy could be the development of a ligand that may mask the peculiar binding motif of Ag85B to block the adhesion. This strategy has been considered against other bacteria and with different purposes but never applied to mycobacteria, becoming a valid antivirulence option to stop the infection at its onset [62,63,64].

### 2.3. Iron Acquisition

Iron is a vital nutrient for most organisms, including the pathogenic *Mab*. It is not only a cofactor of different enzymes but also involved in several other mechanisms including the expression of virulence factors [65,66].

Iron homeostasis is a highly regulated process since the intracellular concentration of this metal can be toxic to the organisms [67]. *Mab* survival depends on iron due to its involvement in different mechanisms such as energy production, DNA replication, and transcriptional regulation. Furthermore, it is implicated in *Mab* capacity to establish and maintain the infection [65,66,68]. *Mab*’s ability to impair the phagosome maturation also depends on iron acquisition processes [66,69]. Nevertheless, the availability of free iron present in host cells is limited. It mainly exists in complex forms, bound to heme-containing proteins, transferrin, lactoferrin, or ferritin [70]. Consequently, all intracellular pathogens must possess the capacity to uptake the iron in a poor environment such as *Mab*’s reproductive niche, to reproduce and establish the infection [66].

Bacteria have evolved sophisticated mechanisms to compete for the host iron, expressing virulence factors that may induce damaging effects [67]. Mycobacteria developed a variety of mechanisms to sequester iron, including the production of mycobactins that are small organic siderophores chelating the ferric iron (Fe^3+^) [66,67]. This is due to the high affinity of mycobactins for Fe^3+^, which commonly exceeds that of the host’s molecules [65,66,67].

The mycobactins are salicylate derivatives that can be classified as lipophilic membrane-associated mycobactins or hydrophilic secreted carboxy-mycobactins [66]. Both are synthetized by the same biosynthetic pathway, and then secreted via specialized secretion systems [71]. The enzymes involved in this pathway are codified by two gene clusters identified in *Mtb.* The *mbtA-J* cluster codifies the enzymes catalysing the biosynthesis of the mycobactin core, while the *mbtK-N* cluster encodes enzymes that add an aliphatic tail specific to membrane-associated mycobactins [72]. The corresponding clusters in *Mab,* from MAB_2119c to MAB_2124, and MAB_2245 to MAB_2250, are represented in Figure 2 [71]. The first enzyme involved in mycobactin production is the salicylate synthase SaS, translated by MAB_2245 and recently well-characterized as an attractive drug target [66,67,73]. In addition, fast-growing mycobacteria synthetize the exochelin, a pentapeptide derivative that is secreted in conditions of iron starvation with the same purpose as mycobactins [70,74]. Ultimately, *Mab* uses the mycobactins as the primary source of soluble iron [67].

The iron uptake also involves the membrane protein complexes MmpLs and ESX-3 that play an important role in the iron acquisition [71,73]. As previously mentioned, ESX-3 is highly preserved in all mycobacterial species, which is consistent with its role in this fundamental pathway [50,73,75].

Effective inhibitors of the mycobactins biosynthesis have been designed to target *Mab* SaS, while other experiments of MAB_2248 deletion, homologous to *M. tuberculosis*’ MbtE, showed their importance for the correct synthesis of mycobactins, thus *Mab* establishment of infection [67,69,74,75].

In addition to the design of new inhibitors targeting enzymes involved in mycobactin biosynthesis, a different strategy was developed to block *Mab* siderophores. The employment of Gallium(III)-based compounds may impair the iron acquisition of different pathogens, *Mab* included. The gallium nitrate Ga(NO_3_)_3_ showed good activity as a competitor of the Fe^3+^ for the binding with the mycobactins, as well as a MICs of 16 µg/mL range [76]. This is due to the similar chemical and physical properties of these two ions. Moreover, the Ga(NO_3_)_3_ showed a synergistic effect with the Ga porphyrin GaPP (showed in Figure 3) in iron deficiency conditions, decreasing the MIC of this combination to 0.06 µg/mL and reducing the bacterial load of 3 logs on a *Mab* infected mice model after 5 days of treatment [76].

Recent studies elucidated the potentiality of encapsulating Ga-based compounds into liquid crystal nanoparticles (LCNP) as delivery systems to improve their delivery and effect [77]. LCNP are cubic or 3D hexagonal-shaped structures formed by self-assembled amphiphilic lipids that possess water channels conferring high solubility to the structure. Thanks to their features LCNP shows an efficient permeation of biological barriers [77,78]. Indeed, upon macrophages’ endocytosis, they fuse with *Mab*’s cell membrane releasing their cargo directly into the pathogen. This was confirmed by a decreased MIC of the GaPP/LCNP of 16 folds with to respect the GaPP only [77]. This strategy integrated with the enhancement of the nanoparticle delivery system opens new possibilities to *Mab* therapies, with the advantages of a low dose, low cytotoxicity, and a focused system against bacterial targets.

### 2.4. Other Virulence Factors

Other factors, such as proteins and genetic elements, may also be associated with *Mab* virulence.

Lsr2 is a histone-like protein that plays a major role in transcriptional regulation and DNA damage protection [79]. Higher levels of Lsr2 were observed in R morphotypes compared to S. Lsr2 knockouts do not show changes in their GPL profile, so the absence of Lsr2 in R morphotype results in the increased susceptibility to reactive oxygen intermediates and reduced survival within macrophages, zebrafish, and murine models [79,80]. Recent studies proved that this protein regulates the transcription of different genes involved in antibiotic resistance and host adaptation in both S and R strains, differently [80]. Therefore, Lsr2 is considered a promising target for drug development against mycobacteria [81,82]. The only example of targeting Lsr2 is related to *Mtb*. The zafirlukast showed an impairment of Lsr2 functions by blocking the DNA complexation [83]. Despite the lack of other examples, probably due to the difficult development of drugs that could bind this protein on a specific motif, this strategy remains unexplored against *Mab*, opening new possibilities to the research.

A two-component system that is important for *Mab* virulence is MtrAB, a complex of two proteins involved in cell division [84]. The lack of one or both mtrA and mtrB genes not only induces a misregulation of cell division but also increases the cell envelope permeabilization [85]. This data further confirms that MtrAB is involved in drug resistance in mycobacterial species [86]. Therefore, the inhibition of the catalytic component of the sensor kinase MtrB, which regulates the gene expression modulator MtrA by phosphorylation, could be a possible strategy to block the mycobacterial adaptation to the environment, including replication within macrophages [84]. Recently, it was found that ΔMtrA, ΔMtrB *Mab* strains or harboring both truncations or the Tyr102Cys mutation on MtrA, showed far greater sensitivity to several drugs than wild-type *Mab*, confirming the possibility of working on this regulator to fight this bacterium [84].

Another two-component system, PhoPR, is implicated in *Mab* virulence. The deletion of the genes codifying for the two proteins PhoP and PhoR produces a hypervirulent strain. However, the mechanisms behind this process are still unclear [87]. A deeper comprehension of the molecular mechanisms regulated by this system would provide new insight into *Mab* virulence, possibly opening a new path to drug design.

### 2.5. Biofilm and Implication

Studies in infected lung tissues of patients with CF and non-CF have recently reinforced the theory that extracellular MABSC biofilm may take place during human pulmonary infections [88]. So, it is important to understand the main features of MABSC biofilm to stop and prevent its formation.

The biofilm is a bacterial complex structure that involves numerous virulence factors during all the stages of its development. According to earlier studies, the *Mab* S morphotype seemed to be associated with the formation of biofilm in aberrant lung airways, while the spontaneous loss of GPL on the surface causes a morphological shift to the invasive R type resulting in inflammation and invasive illness [89,90]. The R variant was shown to be more flexible under both normal and shear stresses, which implies that GPL probably influences the mechanical characteristics of *Mab* biofilm [91]. Nevertheless, it was recently proved that each morphology variant can form biofilm aggregates. For example, the hyper-aggregative R morphotype can establish a biofilm resistant to acidic pH, hydrogen peroxide, and antibiotics [71]. Patients may be colonized with one or both *Mab* variants, without limitation to the biofilm formation or macrophage survival [92].

The biofilm formation starts with bacterial adhesion to the substrate, followed by cell aggregation. This stage is characterized by the upregulation of membrane transport [88]. The formation of the different cell layers, sessile cell growth, and matrix production are typical of the maturation stage, characterized by a switch in gene expression and the increased production of chaperonins [84]. In the last stage of biofilm formation, its density decreases, and the dispersal of cells induces the formation of a new biofilm in a different site [92,93].

Bacterial cells exist in various metabolic states within the biofilm structure [88]. This includes a non-replicating state where cells can live in restrictive conditions of oxygen and nutrients, becoming less vulnerable to physicochemical stress and antibiotics which mostly target replicating cells [94]. Both replicating and non-replicating cells modulate the infection persistency and the colonization of the respiratory epithelium [88]. Treatments suggested by the guidelines may not be enough to eradicate *Mab* within the biofilm, since the external matrix (ECM) is impenetrable to most drugs [95]. This envelope is composed of several elements with both structural and functional roles [95]. Lipids, carbohydrates, proteins, and extracellular DNA (eDNA) are primary examples of the secreted substances forming the ECM. Lipids such as mycolyl-diacylglycerol, mycolic acids, and GPL are considered virulence factors playing a major role in the formation of the biofilm. Moreover, the biofilm resistance to drugs may also be due to other factors. For instance, horizontal gene transfer (HGT), is encouraged by the bacteria’s great closeness in this structure. This mechanism plays a general key role in bacterial survival by expressing acquired efflux pumps or other elements that may confer resistance to antibiotics [96]. HGT inter- and intra-species can also explain the high frequency of mutations that may contribute to the antimicrobial resistance of cells in the biofilm [96].

New therapies are in development to prevent or disrupt this structure. For instance, a universal family of bacterial proteins that bind the eDNA, known as DNABII, has recently been identified as a target for biofilm disruption. Monoclonal antibodies targeting DNABII induced the collapse of *P. aeruginosa*, *S. aureus*, *Burkholderia cenocepcia,* and *Moraxella catarrhalis* biofilms both in vitro and ex vivo in two different animal models [97]. This strategy was successful also against *Mab*, demonstrating the biofilm disruption in vitro and the complete infection clearance in the murine model [98].

An alternative approach is the use of DNases to directly degrade eDNA. Studies on *Mycobacterium fortuitum* and *M. chelonae* demonstrated that the biofilm volume decreases in a dose-dependent manner upon treatment with DNase I [99].

Both strategies result in the release of bacterial cells, making them vulnerable to antibiotics and immune cell attacks.

## 3. Conclusions

Many aspects of metabolism and virulence of pathogenic and opportunistic mycobacteria are still unknown but different researchers are focused on uncovering them. The increased importance of *Mab* as an opportunistic pathogen of fragile individuals pushed different efforts in this direction. Despite the lack of drugs conveniently designed to fight *Mab* infections, repurposed treatments are available. Nevertheless, they have a limited efficacy, mostly for the intrinsic resistance of this bacterium to many drugs. So, the integration of new antimicrobials targeting the *Mab*’s virulence factors into the actual regimen could be of help in shortening the therapeutic time and improving the efficacy. This review provides an overview of the actual knowledge of *Mab* virulence factors and the few ongoing studies to block them. Moreover, it critically suggests other possible elements that could be exploited as targets to effectively block *Mab* virulence and replication in the infection site.

## Figures and Tables

**Figure 1 ijms-26-03247-f001:**
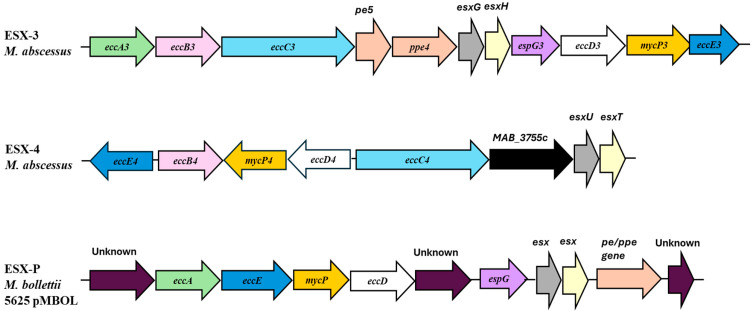
ESX-3, ESX-4, and ESX-P secretion systems in *Mab*; a new ESX system encoded on a plasmid, ESX-P, has been reported in a clinical isolate of *M. bolletii*.

**Figure 2 ijms-26-03247-f002:**
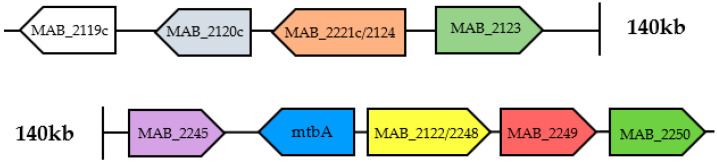
Cluster of the genes codifying for the enzymes involved in mycobactin biosynthesis.

**Figure 3 ijms-26-03247-f003:**
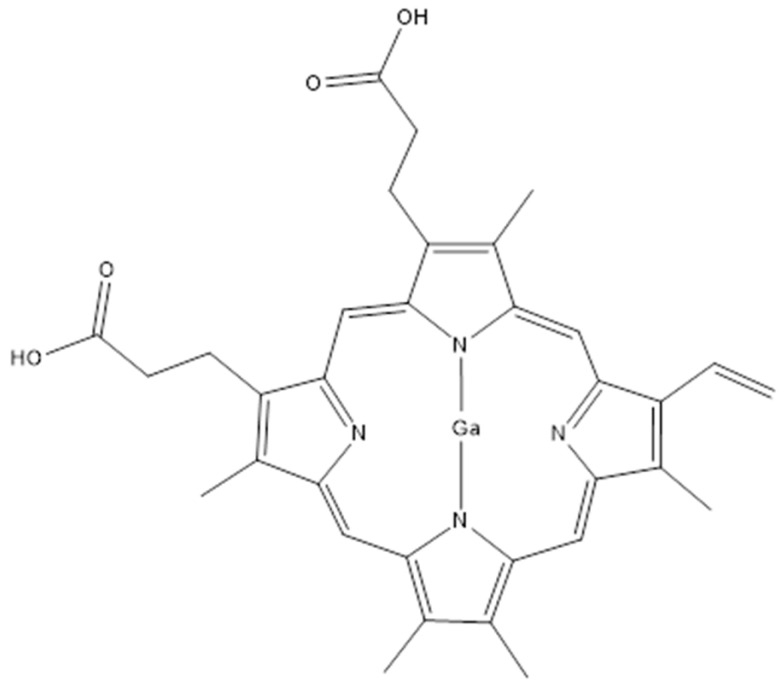
Structure of the gallium porphyrin GaPP.

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
