# Peer review of "Mycobacterium abscessus Virulence Factors: An Overview of Un-Explored Therapeutic Options"

_ijms, 2025, doi:10.3390/ijms26073247_

Round 1
Reviewer 1 Report
Comments and Suggestions for Authors
There is a clear need for new and improved drug regimes for treatment of M. abscessus infection, current treatments are lengthy and the rate of success is low in many cases.
In this manuscript, Cocorullo et al. summarize the main virulence factors of M. abscessus and propose that the development of inhibitors against these virulence factors might shorten treatment and increase success rates against this pathogen. The authors have made a thorough review of the literature and enumerated many of the recently discovered virulence factors of M. abscessus. However, the manuscript would benefit from a careful read-through for grammar and syntax issues by a native English speaker.
It would be a great addition if the authors could mention and summarize if this strategy (targeting virulence factors) has been used for other pathogens and if it has been successful or not in those cases. This would make the whole point of the manuscript more relevant and not a mere review of virulence factors.
In addition, please make sure the citations are correct. Just to name a few mistakes I found:
Citation 83 does not refer to MtrA
Citation 86 does not refer to PhoPR
Other comments
Replace the word “anyway” throughout the text, it is too informal.
Italicize Mab throughout the text.
LINE 99: Surface, not surphase.
LINE 110: Post-translational is used for modifications on proteins, it does not apply to GPLs.
LINE 114: What do the authors mean by “the S morphotype remains silent”?
LINE 136: Do the authors mean “bacterial clumps”? Same line, do these clumps (or clamps) engulf the phagosome or vice versa?
LINE 163: What are I-mannosides?
LINE 300: Attached, not attacked.
LINE 339: Please give a better description of the genomic cluster. Genes coding for mycobactin synthesis are located in two separate clusters (MAB_2119c to MAB_2124, and MAB_2245 to MAB_2250). In line with this, please improve Figure 2, use one or two light colors so the gene number and name can be seen clearly.
LINE 367: Encapsulating instead of incapsulating.
LINE 368: Change the title. Maybe “Other virulence factors”.
LINE 411: Biofilm section. This section is merely descriptive. Which of the virulence factors mentioned could be a target? The authors could at least speculate on different treatments/targets. For example, a recent publication describes the use of monoclonal antibodies against bacterial DNABII proteins to disrupt biofilms with success (Jurcizek et al., 2025).
On the same section LINES 445 to 450. It is true that horizontal gene transfer is increased when bacteria are closer. This could help transfer genes that confer resistance but this mechanism does not induce mutations on drug targets as stated by the authors.
Comments on the Quality of English LanguageThe manuscript would benefit from a careful read-through for grammar and syntax issues by a native English speaker.
Author Response
First of all, we want to thank the reviewer for the care in revisioning our manuscript.
Comments: -It would be a great addition if the authors could mention and summarize if this strategy (targeting virulence factors) has been used for other pathogens and if it has been successful or not in those cases. This would make the whole point of the manuscript more relevant and not a mere review of virulence factors.
In addition, please make sure the citations are correct. Just to name a few mistakes I found:
-Citation 83 does not refer to MtrA
-Citation 86 does not refer to PhoPR
Reply: According to the reviewer comments, we added a paragraph in lines 73-77 to underline that some attempts of antivirulence strategy are already in clinical trials for other pathogens such as P. aeruginosa and S. aureus.
We’re sorry for the mistake in citations, we checked the bibliography and it was the only error we found. Citations are now correct.
Comments: -Replace the word “anyway” throughout the text, it is too informal.
-Italicize Mab throughout the text.
-LINE 99: Surface, not surphase.
Reply: we changed the text in agreement with the reviewer suggestions. Thank you for underlining these points.
Comment: LINE 110: Post-translational is used for modifications on proteins, it does not apply to GPLs.
Reply: In agreement with the reviewer, we changed “post-translational” in “post-synthesis” which is definitely more appropriate. If the reviewer has any other suggestion, we’re happy to improve the text according to his ideas.
Comment: -LINE 114: What do the authors mean by “the S morphotype remains silent”?
-LINE 136: Do the authors mean “bacterial clumps”? Same line, do these clumps (or clamps) engulf the phagosome or vice versa?
Reply: We thank the reviewer for underlining these points. We improved the text to clarify the concept in “the S morphotype survives in a poorly acidified phagosome manipulating the macrophage functions to avoid the phagolysosome fusion [27]. In contrast, the R morphotype clumps may grow and survive within the mature phagolysosome by forming granulomas, which lead to the mechanical disruption of this organelle or the formation of autophagic vacuoles, ultimately resulting in macrophage apoptosis”
Comment: LINE 163: What are I-mannosides?
Reply: Sorry we missed a P, it should have been PI-mannosides, however we decided to change the text in “phosphatidylinositol mannosides” to better clarify.
Comment: LINE 300: Attached, not attacked.
Reply: corrected
Comment: LINE 339: Please give a better description of the genomic cluster. Genes coding for mycobactin synthesis are located in two separate clusters (MAB_2119c to MAB_2124, and MAB_2245 to MAB_2250). In line with this, please improve Figure 2, use one or two light colors so the gene number and name can be seen clearly.
Reply: we improved the text adding a description of the two clusters of genes.
“The enzymes involved in this pathway are codified by two gene clusters identified in Mtb. The mbtA-J cluster codify the enzymes catalyzing the biosynthesis of the mycobactin core, while the mbtK-N cluster encodes enzymes that add an aliphatic tail specific to membrane-associated mycobactins [72]. The corresponding clusters in Mab, from MAB_2119c to MAB_2124, and MAB_2245 to MAB_2250, are represented in Figure 2.”
Comments -LINE 367: Encapsulating instead of incapsulating.
-LINE 368: Change the title. Maybe “Other virulence factors”.
Reply: we changed the text according to the reviewers’ suggestions.
Comment: LINE 411: Biofilm section. This section is merely descriptive. Which of the virulence factors mentioned could be a target? The authors could at least speculate on different treatments/targets. For example, a recent publication describes the use of monoclonal antibodies against bacterial DNABII proteins to disrupt biofilms with success (Jurcizek et al., 2025).
Reply: We thank the reviewer for his very useful comment, we improved the manuscript according to his suggestion adding the following paragraph:
“New therapies are in development to prevent or disrupt this structure. For instance, a universal family of bacterial proteins that bind the eDNA, known as DNABII, has recently been identified as a target for the biofilm disruption. Monoclonal antibodies targeting DNABII induced the collapse of P. aeruginosa, S. aureus, Burkholderia cenocepcia and Morax-ella catarrhalis biofilms both in vitro and ex vivo in two different animal models [98]. This strategy was successful also against Mab, demonstrating the biofilm disruption in vitro and the complete infection clearance in murine model [99].
An alternative approach is the use of DNases to directly degrade eDNA. Studies on Mycobacterium fortuitum and M. chelonae demonstrated that the biofilm volume decreases in a dose dependent manner upon treatment with DNase I [100].
Both strategies result in the release of bacterial cells, making them vulnerable to anti-biotics and immune cells attack.”
Comment: On the same section LINES 445 to 450. It is true that horizontal gene transfer is increased when bacteria are closer. This could help transfer genes that confer resistance but this mechanism does not induce mutations on drug targets as stated by the authors.
Reply: We changed the text in: This mechanism plays a general key role in bacterial survival by expressing acquired efflux pumps or other elements that may confer resistance to antibiotics.

Reviewer 2 Report
Comments and Suggestions for Authors
The authors of this review manuscript about the virulence factors of Mycobacterium abscessus (Mab), an important human pathogen in patients with cystic fibrosis and other immune deficient patients, reviewed the recent literature on the virulence factors of Mab and their associations with pathology of Mab infection in the context of literature on the virulence of Mycobacterium tuberculosis (Mtb). The authors have made efforts to include all the relevant literature and discussed important topics in this context in the physiology and pathology of Mab infection. The information described in this review article is informative and interesting. The only one small concern is that the word “anyway” appeared in the text several times, and it could be replaced by other words with the same meaning or deleted.
Author Response
We want to thank the reviewer for the time spent in reading and reviewing our manuscript. We changed "anyway" or deleted it according the valuable reviewer suggestion.
Thank you, again.
Round 2
Reviewer 1 Report
Comments and Suggestions for Authors
The manuscript has been improved and all my comments have been addressed.
Just a minor comment:
LINE 120: Granulomas are very characteristic formations where a group of immune cells surround mycobacteria so is not the correct word here. Please replace by a more appropriate word (or remove "by forming granulomas" altogether).
Author Response
We want to thank again the reviewer for the care and meticulousness of his work. In agreement with the underlined point, we changed the text in:
"In contrast, the R morphotype clumps may survive within the mature phagolysosome as growing aggregates, which lead to the mechanical disruption of this organelle or the formation of autophagic vacuoles, ultimately resulting in macrophage apoptosis."
We definitely appreciated all the care for our work and the previous suggestions that improved this review.